# Trend of Antihypertensive Medicine Use in the Baltic States between 2008 and 2018: A Retrospective Cross-National Comparison

Indre Treciokiene [1,2,*], Nomeda Bratcikoviene [3,4], Jolanta Gulbinovic [5], Bjorn Wettermark [2,6] and Katja Taxis [1]

1  Department of PharmacoTherapy, Epidemiology & Economics, Faculty of Science and Engineering, University of Groningen, 9713 AV Groningen, The Netherlands; k.taxis@rug.nl

2  Pharmacy Center, Institute of Biomedical Science, Faculty of Medicine, Vilnius University, 03101 Vilnius, Lithuania; bjorn.wettermark@farmaci.uu.se

3  Department of Mathematical Statistics, Faculty of Fundamental Sciences, Vilnius Tech, 10223 Vilnius, Lithuania; nomeda.bratcikoviene@vgtu.lt

4  Department of Human and Medical Genetics, Institute of Biomedical Science, Faculty of Medicine, Vilnius University, 03101 Vilnius, Lithuania

5  Department of Pathology, Forensic Medicine and Pharmacology, Institute of Biomedical Science, Faculty of Medicine, Vilnius University, 03101 Vilnius, Lithuania; jolanta.gulbinovic@mf.vu.lt

6  Department of Pharmacy, Faculty of Pharmacy, Uppsala University, 75105 Uppsala, Sweden

*  Correspondence: i.treciokiene@rug.nl

**Abstract:** High blood pressure is a major risk factor contributing to death and disability rates in the Baltic states. The aim of this study was to compare the utilization of antihypertensive medicines in Estonia, Latvia and Lithuania from 2008 to 2018. In this retrospective cross-national comparison, nationally representative wholesale data from the IQVIA National Retail Audit were analyzed. The utilization of inhibitors of the renin–angiotensin system, beta blockers, calcium channel blockers, diuretics and centrally acting antihypertensives by Defined Daily Doses per 1000 inhabitants and day (DDD/TID) was used to calculate utilization. Time series analysis was used to analyze trends. The utilization increased annually by 10.88, 8.04 and 6.42 DDD/TID in Estonia, Latvia and Lithuania, respectively, from 2008. The utilization of antihypertensive drugs in 2018 was 372, 267 and 379.5 DDD/TID, respectively. Inhibitors of the renin–angiotensin system were the most commonly used class in 2008 and 2018. From 2008, the utilization of beta blockers and fixed-dose combinations including renin–angiotensin system inhibitors increased substantially, while that of calcium channel blockers decreased. Country-specific utilization trends were noted; e.g., the utilization of centrally acting antihypertensives was 30.9 DDD/TID in Lithuania compared to 3.01 DDD/TID in Estonia and 16.17 DDD/TID in Latvia. The use of antihypertensive medicines increased over the study period, but the trends for the different drug classes differed between countries.

**Keywords:** drug utilization; antihypertensives; Baltic states

## 1. Introduction

Hypertension is one of the leading risk factors in terms of attributable disability-adjusted life years at the global level [1]. The initiation and continuation of antihypertensive drug therapy are, next to lifestyle changes, the most important measures for blood pressure control to reduce mortality and morbidity from cardiovascular disease (CVD), as outlined in international guidelines [2,3]. There is substantial evidence that antihypertensive drug treatment reduces the risk of cardiovascular morbidity and mortality [4] and is important to improve the early diagnosis and treatment of hypertension.

The number of people with hypertension has increased significantly in recent decades, partly due to the growing and ageing population [5]. Hypertension treatment has evolved over several decades, driven by the development of various antihypertensive medication

classes [6]. Antihypertensive drug treatment has changed because of the introduction of newer drug classes, drug combinations and subsequent changes in evidence and guideline recommendations [7]. On the other hand, antihypertensives are also used to treat other CVDs and other diseases as well. For example, beta blockers are effective at reducing both exercise-induced angina and asymptomatic ischemic episodes [8], and diuretics are used in pulmonary edema and heart failure among other indications [9]. The use of antihypertensives for indications other than hypertension makes these drug classes heterogeneous in terms of indication.

The Baltic countries are similar in terms of geography, size, economic structure [10] and general population characteristics; e.g., the distribution of people over 65 years old is 19% in Lithuania and Estonia and 20% in Latvia [11–13]. However, some differences in health status have been observed between the countries. Cardiovascular death rates are similar in Lithuania and Latvia, constituting 53% of total deaths compared to 34% in Estonia [12,14,15] and 24% in Denmark and France [16]. The age-standardized death rate for CVD in Lithuania and Latvia is one of the highest in the European Union [17]. There are some differences in the organization of the healthcare systems between the Baltic countries. In Estonia and Lithuania, the national health insurance systems are public and cover approximately 95% [18] and 97% [19] of the population, respectively. The Latvian healthcare system is characterized by tax-financed statutory healthcare provision, a purchaser–provider split and a mix of public and private providers. Approximately one-third of the Latvian employed population is enrolled into a voluntary health insurance paid for by the employers [20]. In the last decade, all three countries operated price regulations for medicines at the level of reimbursement of those (partially) funded by the state and at the level of wholesale and retail prices [21]. Other instruments which influence drug utilization include the lists of medicines eligible to be reimbursed, which were also used in all three countries. In all Baltic countries, large out-of-pocket payments have been observed [13].

Studies from a range of countries showed that the use of antihypertensive medicines has increased steadily over recent decades [22–26]. The decrease in CVD mortality in Europe that has been observed in recent years is attributed to active preventive practices and proper treatment [27,28]. Only a limited number of studies on the utilization of antihypertensive drugs have been performed in the Baltic countries. In 2017, the utilization of cardiovascular medicines and cardiovascular mortality were compared between Lithuania, Sweden and Norway [29]. This study showed that the utilization of cardiovascular medicines was associated with a decline in cardiovascular age-standardized death rate. Another report revealed that the total consumption of cardiovascular drugs increased annually by 16.7% in Lithuania from 2003 to 2012 [30]. The Estonian, Latvian and Lithuanian State Agencies of Medicines published shared reports on Baltic Statistics on Medicines [31], which included sales to both pharmacies and hospitals, yet the trends over a longer period were not investigated in detail. In summary, little is known about the trends in the utilization of antihypertensives, despite very high levels of cardiovascular mortality in the Baltic countries.

We have highlighted the similarities and differences of three Baltic countries. Comparing trends in the context of similarities and differences in population, healthcare systems and availability and affordability of antihypertensive medicines is the basis to inform strategies for better management of hypertension and other cardiovascular diseases. The aim of this study was to compare the utilization of antihypertensive medicines in Estonia, Latvia and Lithuania from 2008 to 2018.

Ethical approval was not needed for this research as the collected data do not contain individual health data.

## 2. Results

### 2.1. Utilization of Antihypertensives

The total utilization of antihypertensive medicines in 2018 was 372 DDD/TID in Estonia, 267 DDD/TID in Latvia and 379.5 DDD/TID in Lithuania. The overall trend is presented in Figure 1.

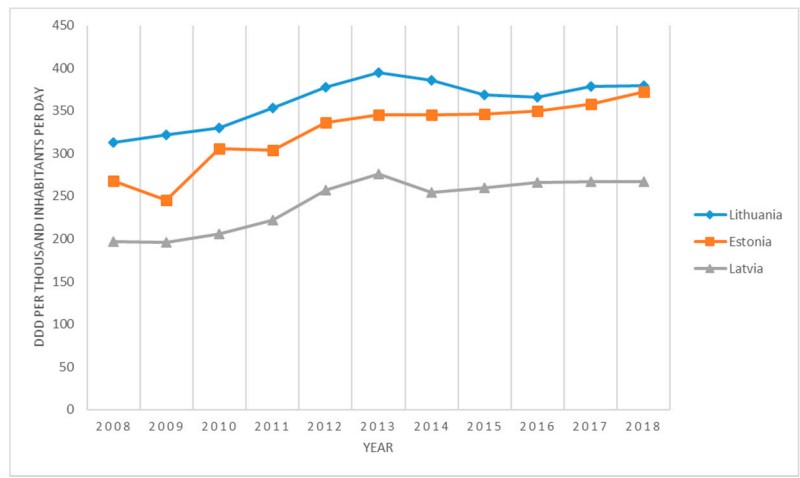

**Figure 1.** Overall trend of utilization of antihypertensive medicines from 2008 to 2018.

Between 2008 and 2018, the use of antihypertensives increased annually by 10.88 DDD/TID in Estonia (95% CI: 7.13–14.63, $p = 0.0001$, $R^2 = 0.827$), 8.04 DDD/TID in Latvia (95% CI: 4.57–11.52, $p = 0.0005$, $R^2 = 0.753$) and 6.42 DDD/TID in Lithuania (95% CI: 2.44–10.41, $p = 0.005$, $R^2 = 0.597$).

### 2.2. Utilization by Pharmacological Group

In 2008, renin–angiotensin system (RAS) inhibitors (C09) were the most commonly used antihypertensive drugs in all Baltic countries with 146, 182 and 105 DDD/TID in Estonia, Latvia and Lithuania, respectively (Figure 2).

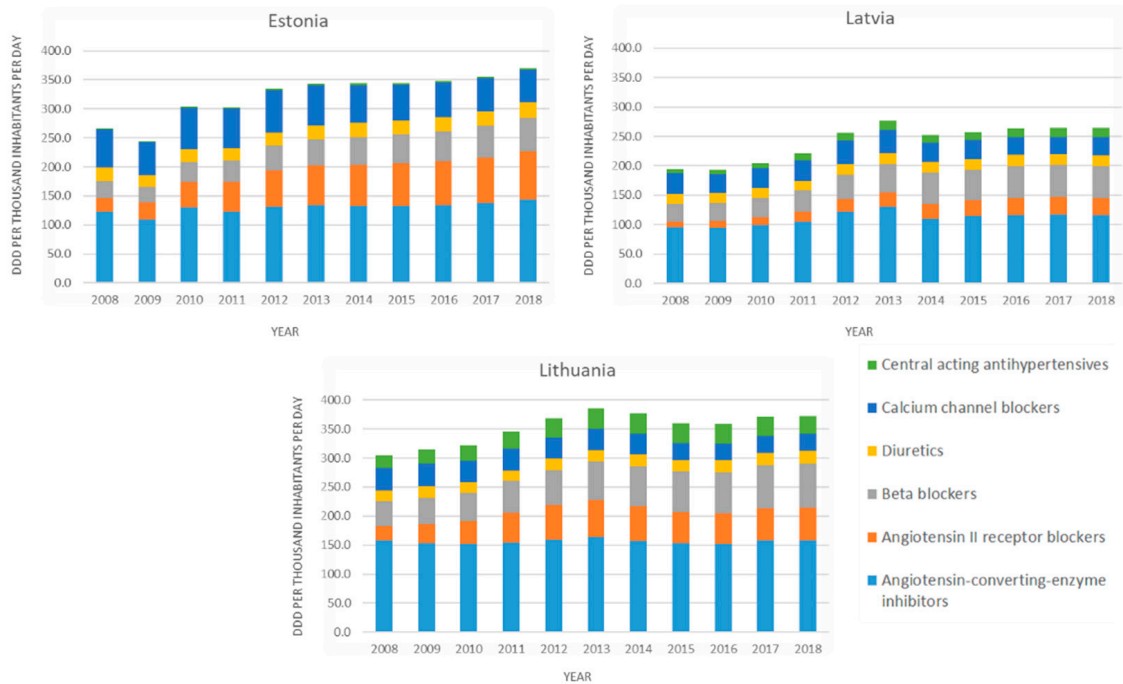

**Figure 2.** Utilization of antihypertensive medicines by drug class in Estonia, Latvia and Lithuania.

The utilization of RAS inhibitors increased significantly in all Baltic countries during the study period. The increase was mostly caused by RAS combinations (Figure 3).

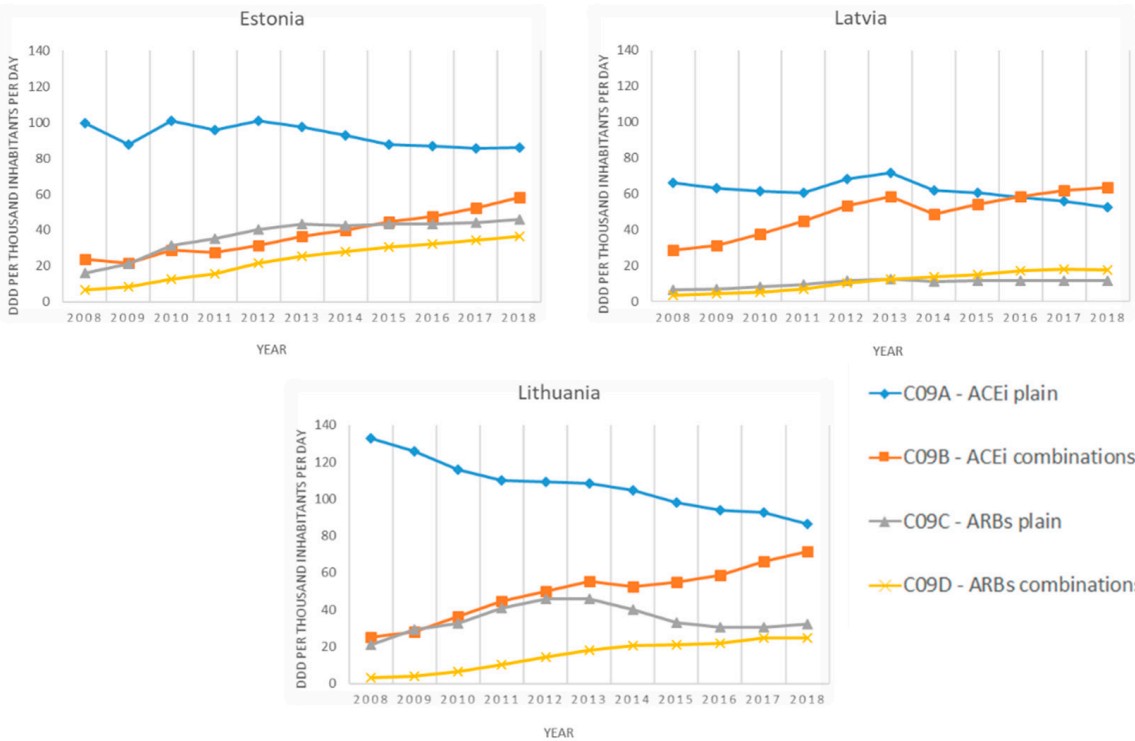

**Figure 3.** Trend of utilization of RAS inhibitors in the Baltic states from 2008 to 2018.

The second most commonly used group in 2008 was calcium channel blockers (C08) in Estonia and Latvia and beta blockers (C07) in Lithuania. Utilization of beta blockers increased in all three Baltic countries, making beta blockers the group with the largest increase in the 11-year period. During the same period, the utilization of calcium channel blockers decreased. The utilization of medicines by drug class is presented in Table 1.

**Table 1.** Trend of utilization of antihypertensive medicines by drug class (ATC).

| | | Country | DDD/TID in 2008 | DDD/TID in 2018 | Mean Annual Increase in DDD/TID | 95% CI of Mean Annual Increase | *p*-Value, Time Series Analysis * | $R^2$ |
|---|---|---|---|---|---|---|---|---|
| Antiadrenergic agents centrally acting C02AC | | Estonia | 1.25 | 3.01 | 0.17 | 0.12–0.23 | <0.0001 | 0.839 |
| | | Latvia | 7.1 | 16.17 | 1.04 | 0.34–1.75 | 0.0086 | 0.554 |
| | | Lithuania | 21.85 | 30.9 | 0.89 | 0.59–1.19 | <0.0001 | 0.832 |
| Diuretics C03 | | Estonia | 22.76 | 27.17 | 0.56 | 0.33; 0.77 | 0.0003 | 0.772 |
| | | Latvia | 17.01 | 17.54 | 0.2 | 0.02; 0.4 | 0.034 | 0.407 |
| | | Lithuania | 18.97 | 22.71 | 0.39 | 0.22; 0.55 | 0.0005 | 0.755 |
| Beta blockers C07 | Plain | Estonia | 29.5 | 57.4 | 2.98 | 2.55; 3.42 | <0.0001 | 0.964 |
| | | Latvia | 29.1 | 50.7 | 2.63 | 1.97; 3.30 | <0.0001 | 0.899 |
| | | Lithuania | 42.4 | 69.4 | 2.77 | 2.27; 3.27 | <0.0001 | 0.946 |
| | Combinations with diuretics | Estonia | 0.1 [a] | 0 | - | - | - | - |
| | | Latvia | 1.3 | 3.4 | 0.26 | 0.21; 0.30 | <0.0001 | 0.9489 |
| | | Lithuania | 0.1 | 5.1 | 0.63 | 0.48; 0.79 | <0.0001 | 0.901 |
| | Other combinations | Estonia | 0.0 | 0.0 | - | - | - | - |
| | | Latvia | 0.0 | 0.6 [d] | - | - | - | - |
| | | Lithuania | 0.0 | 0.5 [e] | - | - | - | - |

**Table 1.** *Cont.*

| | | Country | DDD/TID in 2008 | DDD/TID in 2018 | Mean Annual Increase in DDD/TID | 95% CI of Mean Annual Increase | *p*-Value, Time Series Analysis * | R² |
|---|---|---|---|---|---|---|---|---|
| Calcium channel blockers C08 | Plain | Estonia | 66.6 | 56.0 | −1.00 | −2.16; 0.17 | 0.0856 | 0.293 |
| | | Latvia | 34.7 | 29.6 | −0.67 | −1.45; 0.06 | 0.0662 | 0.327 |
| | | Lithuania | 39.7 | 24.1 | −1.72 | −2.25; −1.2 | <0.0001 | 0.859 |
| | Combinations with diuretics | Estonia | 0.0 | 0.0 | - | - | - | - |
| | | Latvia | 0.0 | 0.3 [d] | - | - | - | - |
| | | Lithuania | 0.0 | 1.1 [d] | - | - | - | - |
| | Other | Estonia | 0.0 | 0.0 | - | - | - | - |
| | | Latvia | 0.0 | 1.3 [c] | - | - | - | - |
| | | Lithuania | 0.0 | 3.8 [f] | - | - | - | - |
| Angiotensin-converting enzyme inhibitors C09A, C09B | Plain | Estonia | 99.8 | 85.8 | −1.32 | −2.36; −0.30 | 0.0172 | 0.485 |
| | | Latvia | 66.3 | 52.4 | −1.06 | −2.02; −0.09 | 0.0355 | 0.404 |
| | | Lithuania | 133.1 | 86.7 | −4.18 | −4.84; −3.51 | <0.0001 | 0.957 |
| | Combinations with diuretics | Estonia | 23.4 | 33.0 | 1.00 | 0.69; 1.32 | <0.0001 | 0.851 |
| | | Latvia | 23.6 | 31.6 | 0.50 | −0.09; 1.09 | 0.088 | 0.289 |
| | | Lithuania | 28.6 | 31.2 | 0.77 | 0.57; 0.97 | <0.0001 | 0.893 |
| | Other | Estonia | 0.2 | 25.2 | 2.6 | 2.14; 3.05 | <0.0001 | 0.949 |
| | | Latvia | 0.2 | 32.4 | 2.89 | 2.27; 3.51 | <0.0001 | 0.925 |
| | | Lithuania | 1.5 | 39.7 | 3.54 | 2.86; 4.22 | <0.0001 | 0.940 |
| Angiotensin II receptor blockers C09C, C09D | Plain | Estonia | 16.1 | 46.0 | 2.70 | 1.61; 3.77 | 0.0003 | 0.780 |
| | | Latvia | 6.7 | 11.4 | 0.51 | 0.25; 0.76 | 0.0014 | 0.697 |
| | | Lithuania | 21.1 | 32.3 | 0.31 | −1.40; 2.02 | 0.6917 | 0.018 |
| | Combinations with diuretics | Estonia | 6.6 | 18.5 | 1.15 | 0.61; 1.68 | 0.0009 | 0.724 |
| | | Latvia | 3.6 | 10.9 | 0.83 | 0.61; 1.05 | <0.0001 | 0.891 |
| | | Lithuania | 3.1 | 14.6 | 1.20 | 0.75; 1.64 | 0.0002 | 0.807 |
| | Other | Estonia | 0.0 | 18.1 [b] | - | - | - | - |
| | | Latvia | 0.0 | 6.8 | 0.81 | 0.66; 0.95 | <0.0001 | 0.947 |
| | | Lithuania | 0.0 | 10.1 [b] | - | - | - | - |

* Time series analysis was used to calculate trends over the 11-year period. R² shows the proportion of the variance; the higher R² values (closer to 1) represent smaller differences between the observed data and the fitted values. A utilization period shorter than 10 years was too short to measure the trend. [a] Sold in 2008–2011 and 2013; [b] introduced in 2010; [c] introduced in 2012; [d] introduced in 2014; [e] introduced in 2015; [f] introduced in 2016.

Utilization of diuretics in monotherapy and fixed combinations increased in all countries. In 2008, the least used groups were centrally acting agents (C02AC) in Estonia and Latvia and plain diuretics in Lithuania. By 2018, utilization of antiadrenergic centrally acting agents (mainly imidazoline receptor agonists moxonidine and rilmenidine) had increased in Latvia and Lithuania by around 1 DDD/TID annually.

Most fixed combinations used in 2008 were combinations with diuretics. During the 11-year period, the utilization of combinations increased mostly due to the increase in combinations not including any diuretics. The total utilization of fixed combinations increased by three to four times in all countries: from 30.3 to 94.9 DDD/TID in Estonia, from 33.7 to 87.1 DDD/TID in Latvia and from 28.2 to 106.6 DDD/TID in Lithuania from 2008 to 2018. The largest increase was observed in the group of combinations containing RAS agents combined with drugs other than diuretics.

Utilization of medicines containing diuretics, both plain and in combinations, increased by 2.70 DDD/TID (95% CI: 1.95–3.45, *p* < 0.0001, R² = 0.88) annually from 52.9 DDD/TID in 2008 to 78.7 DDD/TID in 2018 in Estonia, by 1.82 DDD/TID (95% CI: 0.91–2.74, *p* = 0.0015, R² = 0.694) annually from 50.5 DDD/TID in 2008 to 63.5 DDD/TID in 2018 in Latvia and by 3.10 DDD/TID (95% CI: 2.45–3.74, *p* < 0.0001, R² = 0.929) annually from 45.7 DDD/TID in 2008 to 75.1 DDD/TID in 2018 in Lithuania. The most frequently used combination containing diuretics in 2018 was angiotensin-converting enzyme inhibitors with diuretics (C09B1) with 33 DDD/TID in Estonia, 31.2 DDD/TID in Latvia and 31.6 DDD/TID in Lithuania.

## 3. Discussion

In this large population-based study, complete data from three Baltic countries were used. From the 11-year study period, there were three main findings.

Firstly, the utilization of antihypertensive medicines increased significantly in Estonia, Latvia and Lithuania during the study period, approaching the utilization rate in Scandinavian and Western countries. Comparison of our results to studies analyzing aggregated data on the utilization of antihypertensives without information on indication showed similarities and differences. The utilization rate was 379 DDD/TID in Denmark in 2015 [25], 476 DDD/TID in Croatia in 2016 [26], 299 DDD/TID in Spain in 2012 [32] and 275 DDD/TID in Norway in 2016 [33]. The annual increase in the utilization of antihypertensive medication is, thus, in line with findings from other countries [25,26,34,35]. In those studies, the increase was mostly attributed to the implementation of hypertension and CVD management guidelines.

Secondly, our study showed similar trends in the utilization of some antihypertensive medicines in all three Baltic countries. RAS inhibitors were the most frequently used class of drugs over the whole study period in all countries. In 2008, plain ACE inhibitors were the main utilized RAS inhibitors. This is in line with the European Society of Cardiology and European Society of Hypertension's recommendations of 2003 to prescribe monotherapy as an initial treatment. If blood pressure was not controlled with monotherapy, the guidelines recommended moving to combination therapy [36]. The 2013 ESH/ESC Guidelines [37] and 2018 ESH/ESC Guidelines [2] favor the initiation of treatment with combinations of two antihypertensive drugs in a single pill. It seems that those recommendations have been followed, as the utilization of plain ACE inhibitors has decreased over the years and the utilization of plain diuretics remained moderate, while the use of their fixed combinations has increased. The utilization of ARBs increased as well, possibly due to the expiration of Losartan's patent in 2009. The utilization of beta blockers showed the highest annual increase during the study period in all Baltic states. The same increase in beta blockers' use was detected in other countries, possibly showing an increase in use in a wide range of cardiovascular diseases [25]. At least in Lithuania, the prevalence of CVD tended to increase until 2014 and declined slightly in 2015–2017, with ischemic heart disease and stroke being the main diseases [38].

Thirdly, this study also found marked differences between the Baltic states. After gaining independence, the Baltic countries started with very similar pharmaceutical policies, with all three countries using centralized purchase of medicines. The differences in trends over the study period could reflect differences in the development of their healthcare systems. Overall, Latvians utilized fewer antihypertensive medicines than Estonians and Lithuanians did both in 2008 and 2018. Centrally acting antiadrenergic agents were the least used class in Estonia and Latvia. Remarkably, in Lithuania, the utilization of this class was higher than in the other two countries both in 2008 and 2018. Over the whole study period, this class' utilization was higher than that of plain diuretics in Lithuania. Those medicines might be prescribed to treat resistant hypertension, which is associated with a high CV risk [39], or to obese patients as recommended by a few authors [40]. Some differences in the medicines used were also noticed; e.g., in Estonia, hydrochlorothiazide was the most used from low ceiling diuretics, while in Latvia and Lithuania, indapamide use surpassed hydrochlorothiazide use. The reason for this could be similar to Australia's case in 2008, where indapamide was the most expensive and only thiazide was advertised as a single agent [41]. The differences in drug classes and medicines used might have been influenced by political decisions and healthcare system regulations as well as reimbursement regulations [42], differences in the marketing strategies of pharmaceutical companies [43] or other differences in local traditions/cultures of prescribing [44].

The utilization trends of antihypertensive medicines may help provide some suggestions for the improvement of hypertension management. The increased use of fixed combinations of two antihypertensive drugs in a single pill may improve adherence and increase the rate of BP control and is favored by guidelines [2]. However, the guidelines do

not recommend centrally acting agents for hypertension management as there is a lack of large-scale outcome trials with evidence on health outcomes and survival [39,45].

This study's strength is its use of a nationwide comprehensive and comparable data source to describe the trends of utilization of antihypertensive medicines in the Baltic states over a decade. Prescription or dispensing data for the Baltic states have previously been difficult to access for research and analyses. The aggregated data used in this study were collected with the same method for all the Baltic countries, which facilitated the comparisons. For this study, IQVIA data were probably superior to other data sources such as administrative database sets from national drug agencies. Overall, we found a clear pattern of trends. At least in Lithuania, it was shown that the recorded utilization of RAS inhibitors was similar between commercial and administrative databases in 2004–2012 [46].

Some limitations exist within this study. First, we used aggregated data that could not be linked to any personal data, and we therefore did not have any clinical information to verify that the drugs were used for the treatment of hypertension and possibly overestimated the use of drugs to treat hypertension. On the other hand, our aggregated data also cover the medicines paid for out of pocket. This might be important in this study because of the large share of drug spending not covered by reimbursement in the Baltic countries. Second, the utilization assessment discrepancies in Estonia could be due to a lower wholesale data panel coverage as described in the Methods section. Third, some limitations within the ATC/DDD classification may also have had an impact on the results. The classification of combination drugs is a challenge. The DDDs assigned for combination products are based on the main principle of counting the combination as one daily dose, regardless of the number of active ingredients included in the combination [47]. Due to the DDD's limitation, drug utilization seems to decrease if a treatment is changed from two medicines to a combination, while the true use of medicines stays the same. On the other hand, as the combinations increased, the true medicine use increased even more than the data show.

Further research should use data sources including information on diagnoses, prescriptions, patients' use of medicines and treatment outcomes to assess the quality of the treatment. The focus should be on exploring the low utilization of all antihypertensives in Latvia, which might be due to undertreatment, and the high utilization of centrally acting antihypertensives in Lithuania.

## 4. Materials and Methods

This was a cross-national drug utilization study of the sales of antihypertensive medicines in Estonia, Latvia and Lithuania between 2008 and 2018.

### 4.1. Data

Data were generated and obtained under a license agreement from the IQVIA National Retail Audit, Vilnius, Lithuania in January 2020. IQVIA aggregates wholesale data covering comprehensive drug sales to all community pharmacies monthly. The data contain brand and INN names with the dosages, packages and wholesale prices of the medicines distributed to pharmacies, excluding drug sales to hospitals. IQVIA estimated that the average deviation between IQVIA data and real drug utilization is 14.3% in Estonia, 4.7% in Latvia and 0.9% in Lithuania—i.e., the overall coverage of drug sales to community pharmacies is above 85% [48]. This difference in Estonia is due to the lower coverage of wholesalers, but this is accounted for in the data provided. The Anatomical Therapeutic Chemical/Defined Daily Dose (ATC/DDD) methodology was used. Defined daily dose (DDD) data were obtained from the WHO Collaborating Centre for Drug Statistics Methodology, using the ATC/DDD Index 2019 [49].

The following antihypertensive agents were included: antiadrenergic agents, centrally acting agents (C02AC), diuretics (C03), beta receptor blockers (C07 plain and combinations), calcium channel blockers (C08 plain and combinations) and renin–angiotensin system inhibitors (C09A—angiotensin-converting enzyme inhibitors (ACEis) plain; C09B—ACEi

in combinations; C09C—angiotensin II receptor blockers (ARBs) plain; C09D—ARBs in combinations). Other agents have been used as antihypertensives, such as alpha blockers, but were excluded from this study. The results are expressed in DDD per thousand inhabitants per day (DDD/TID). For example, 10 DDD/1000 inhabitants/day for drug A in country X indicates that on average, 1% of the population from country X receives treatment A daily. Utilization was calculated regardless of indication, although some of the drugs have several indications other than hypertension, including heart failure, angina pectoris and arrhythmias.

The number of inhabitants for each studied year was retrieved from national statistics databases: Statistics Estonia [50], Statistics Lithuania [51] and the Central Statistical Bureau of Latvia [52]. The number at the beginning of each year was used for calculation of TID.

*4.2. Statistical Analyses*

Data were analyzed by time series analysis using linear regression to analyze the trends in each country over the 11-year period. The $R^2$ value for testing the fit of the model was calculated. An $R^2$ value of 1 means that a 100% fit between the regression model and the data points is present. Additionally, 95% confidence intervals around the shape and *p*-values of the linear regression were calculated. Statistical analyses were conducted using the Microsoft Data Analysis ToolPak for Excel 2016 version.

**5. Conclusions**

The use of antihypertensive medicines increased in Estonia, Latvia and Lithuania over the study period, but the trends for the different drug classes differed between the countries. Antihypertensive medicines' utilization was lower in Latvia compared to the other two countries. In Lithuania, the utilization of centrally acting agents was higher than in Estonia and Latvia both in 2008 and 2018.

**Author Contributions:** Conceptualization, I.T. and K.T.; methodology, I.T., K.T. and B.W.; formal analysis, I.T.; resources, J.G.; data curation, N.B.; writing—original draft preparation, I.T.; writing—review and editing, J.G., B.W. and K.T.; supervision, K.T. All authors have read and agreed to the published version of the manuscript.

**Funding:** This research received no external funding.

**Institutional Review Board Statement:** Ethical approval was not needed for this research as the collected data do not contain individual health data.

**Informed Consent Statement:** Not applicable.

**Data Availability Statement:** The data presented in this study are available on request from the corresponding author. The data are not publicly available due to restrictions apply to the availability of these data. Data were obtained by the license agreement from IQVIA Baltic and are available upon reasonable request and with the permission of IQVIA Baltic.

**Acknowledgments:** Special thanks are given to the IQVIA Baltic team for providing the data.

**Conflicts of Interest:** The authors declare no conflict of interest.

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
