# Peer review of "Trend of Antihypertensive Medicine Use in the Baltic States between 2008 and 2018: A Retrospective Cross-National Comparison"

_2813-0618, doi:10.3390/pharma1010001_

Round 1

Reviewer 1 Report

This is a well-written report of a population-based utilization study for antihypertension drugs in Latvia, Estonia, and Lithuania. The data and methods are straightforward and appropriate, and the findings clearly presented. I have no major concerns with this article as written, however I do have a few minor comments.

The figures should show the units for the values on the Y axes, either in the figures themselves or at least in the captions. Units in the accompanying text of the article is not sufficient.

Line 134 - cut the word "those"

Line 157 - "increases" should just be "increase"

Line 173 - "outwent" is a relatively obscure word, which I believe generally has a monetary connotation. I would recommend replacing this with "surpassed"

Line 204 - "challenging" should be "challenge"

Section 4.1 feels more like introduction to me than materials and methods. 

Overall this is a high-quality manuscript.

Author Response

Thank you for your thorough review of our manuscript and the comments and suggestions. We have addressed all of those point to point. We have also included the manuscript with the changes marked using the track change function. Two types of changes are not marked to keep the manuscript readable and clear. This is firstly the numbering of the references. Because we moved information from chapter 4.1 to the introduction, the numbers of the references all changed. Secondly, we have updated the figures as detailed below.

The figures should show the units for the values on the Y axes, either in the figures themselves or at least in the captions. Units in the accompanying text of the article is not sufficient.

- Figures updated (lines 101, 112, 117)

Line 134 - cut the word "those"

- Addressed (line 156)

Line 157 - "increases" should just be "increase"

- Addressed (line 181)

Line 173 - "outwent" is a relatively obscure word, which I believe generally has a monetary connotation. I would recommend replacing this with "surpassed"

- Recommendation accepted, word replaced (line 198)

Line 204 - "challenging" should be "challenge"

- Addressed (line 228)

Section 4.1 feels more like introduction to me than materials and methods. 

- Information from section 4.1. has been moved and slightly adapted as part ofthe introduction (lines 54 - 72)

Overall this is a high-quality manuscript.

- Thank you

Reviewer 2 Report

Thank you for your great efforts in this research of the antihypertensive drug utilization trend.

Here are some suggestions that may be necessary for the purpose of improving the manuscript's content:

1. Row 42: What does the "recognition of hypertension" mean?

2. What is the rationale when the trends in utilization of antihypertensives are known? For example, will it contributes to the national drug policy or maybe the drug price and so on.

3. What is the problem statement of this study? Is there any reason to compare among selected only, i,e, Estonia, Latvia and Lithuania ... is it because of the inequalities in the health care systems?

4. Row 141: Kindly include, if there is any justification (mentioned by the other countries) of the annual increase in the utilization of antihypertensive medication.

5. Row 164: It will be helpful to clarify the "inequalities in the health care systems". Is it relate to which state has the least access to medication supply or so on? (This comment can be addressed if only the "Material & Method" appeared before the discussion part.
However, it will be helpful if the findings are discussed by comparing side by side with differences in the organization of the health care systems between the Baltic countries.

6. Row 216: Materials & Methods should be located before Results.

7. Row 277 (Conclusion): Row 277 (Conclusion): Is it possible to state that the differences in the healthcare system, led to the differences in utilization trend among Estonia, Latvia and Lithuania?

Author Response

Thank you for your thorough review of our manuscript and the comments and suggestions. We have addressed all of those point to point. We have also included the manuscript with the changes marked using the track change function. Two types of changes are not marked to keep the manuscript readable and clear. This is firstly the numbering of the references. Because we moved information from chapter 4.1 to the introduction, the numbers of the references all changed. Secondly, we have updated the figures No.1, No.2 and No.3 with the units of both axes.

Does the introduction provide sufficient background and include all relevant references? – must be improved.                     

- We have amended the introduction to include information on similarities and differences of the three Baltic countries; there a reason to compare trends between three countries, as the differences and similarities found might be a solid base to inform further investigations.

  1. Row 42: What does the "recognition of hypertension" mean?

- in this paper it was meant as an early diagnosis, we have rephrased the sentence (line 42)

  1. What is the rationale when the trends in utilization of antihypertensives are known? For example, will it contributes to the national drug policy or maybe the drug price and so on.

- As we have detailed in the introduction, a very limited number of studies have been done in the Baltic countries which provide some general information on the trends. But detailed information on similarities and differences in drug utilization between the countries over a considerable time span are not available. Given the differences and similarities in the health care systems, populations etc between the Baltic countries, our study results are a solid base to inform further investigations. This might also influence the national drug policy. We have amended the introduction section with more details about the Baltic States and the rational of our study to address this comment.  

  1. What is the problem statement of this study? Is there any reason to compare among selected only, i,e, Estonia, Latvia and Lithuania ... is it because of the inequalities in the health care systems?

- Thank you for the comment, we have meant differences in health care systems not inequalities in this study. Comparing trends between the three Baltic countries in the context of similarities and differences in population, health care systems, availability and affordability of antihypertensive medicines is a basis to inform strategies for better management of hypertension and other cardiovascular diseases.

  1. Row 141: Kindly include, if there is any justification (mentioned by the other countries) of the annual increase in the utilization of antihypertensive medication.

- addressed (lines 164-166)

  1. Row 164: It will be helpful to clarify the "inequalities in the health care systems". Is it relate to which state has the least access to medication supply or so on? (This comment can be addressed if only the "Material & Method" appeared before the discussion part.
    However, it will be helpful if the findings are discussed by comparing side by side with differences in the organization of the health care systems between the Baltic countries.

- Sorry, it was our mistake, we meant differences in the developments of the health care systems. Comparing side by side health care differences with drug utilization trends was not the aim of this study. We understand that the „Method“ section after the „Results“ is not common, yet we followed the requirements of MDPI that were presented both in the template and instructions for authors. Nevertheless, we addressed the issue and moved the information about the differences and similarities of the Baltic countries from section 4.1 to the introduction.

  1. Row 216: Materials & Methods should be located before Results.

- as addressed above, we followed the instructions for authors. If this is a substantial deficiency of the paper, we would consult editorial office for the exception.

  1. Row 277 (Conclusion): Row 277 (Conclusion): Is it possible to state that the differences in the healthcare system, led to the differences in utilization trend among Estonia, Latvia and Lithuania?

- We have addressed the possible explanations for the observed differences in drug utilization in the discussion section including possible differences in the health care systems. Whether or not those differences really led to those differences is outside the scope of this paper. Investigating possible correlation/causal relationships requires more sophisticated study methods, because many factors may influence. We address possible follow up studies investigating those aspects also in the discussion section. Therefore, we would like to keep the current conclusion which is based on the actual study results.